# The Roles of Bacteria in Soil Organic Carbon Accumulation under Nitrogen Deposition in *Stipa baicalensis* Steppe

**DOI:** 10.3390/microorganisms8030326

**Published:** 2020-02-26

**Authors:** Jie Qin, Hongmei Liu, Jianning Zhao, Hui Wang, Haifang Zhang, Dianlin Yang, Naiqin Zhang

**Affiliations:** 1Agro-Environmental Protection Institute, Ministry of Agriculture and Rural Affairs; Key Laboratory of Original Agro-Environmental Pollution Prevention and Control, MARA; Tianjin Key Laboratory of Agro-Environment and Agro-Product Safety, Tianjin 300191, China; qinjie@caas.cn (J.Q.); liuhongmei@caas.cn (H.L.); zhaojianning@caas.cn (J.Z.); wanghui03@caas.cn (H.W.); hfzhang12@126.com (H.Z.); 2Department of Ecology and Landscape Architecture, Dezhou University, Dezhou 253023, China

**Keywords:** nitrogen deposition, grassland, soil organic carbon, bacteria, soil properties

## Abstract

Grassland soil organic carbon (SOC) accounts for 15.5% of the SOC in reservoirs of terrestrial carbon (C) and is a major component of the global C cycle. Current and future reactive N deposited on grassland soils may alter biogeochemical processes and soil microbes. Microorganisms perform most of the decomposition on Earth and shift SOC accumulation. However, how variation in the identity and composition of the bacterial community influences SOC is far from clear. The objective of this study is to investigate the responses of SOC concentration to multiple rates of N addition as well as the roles of bacteria in SOC accumulation. We studied SOC storage and bacterial community composition under N addition treatments (0, 1.5, 3.0, 5.0, 10.0, 15.0, 20.0, and 30.0 g N·m^−2^ yr^−1^) in a 6-yr field experiment in a temperate grassland. We determined the soil inorganic nitrogen concentration and pH in a 0–10 cm soil layer. We used high-throughput genetic sequencing to detect bacteria. N addition led to significant increases in the concentrations of SOC. N addition reduced the soil pH but increased the NO_3_-N and NH_4_-N levels. The bacterial diversity was highest under low nitrogen addition. N addition increased the relative abundance of Proteobacteria, and Proteobacteria became the second dominant phylum under high N addition. Structural equation modeling further revealed that soil pH and bacterial community structure have an impact on SOC under N deposition. Nitrogen-regulated SOC is associated with Proteobacteria and Planctomycetes. These findings suggest that N deposition may alter the SOC content, highlighting the importance of understanding changes in the bacterial community for soil nutrients under N deposition.

## 1. Introduction

Atmospheric deposition of reactive nitrogen (N) has significantly increased globally in the last few decades, with the most rapid increase observed in developing countries. Human interference with the global N cycle has far outweighed that with any other major biogeochemical cycle on Earth. Human activities have increased atmospheric N deposition by 3- to 5-fold since the last century [1]. N deposition has increased greatly since the beginning of the industrial revolution. At present, East Asia (mainly China), Western Europe and North America have become three hot spots of global atmospheric N deposition [2]. It is predicted that by 2050, global reactive N deposition will reach 195 Tg N·yr^−1^ [3]. Since the reformation and opening of China, N deposition caused by production activities and air pollution has also been increasing. The N deposition reached 1.0-1.5 g N·m^−2^ in China’s northern grasslands in 2008 [4,5]. Much attention has been given to the extent and impacts of the altered deposition of N. Some research found that increasing N deposition alters plant diversity [6] and soil bacterial diversity [7] and changes the original soil properties, including the C/N balance and pH [8], hence changing the global ecosystem [9].

N deposition is a component of global change that has a considerable impact on belowground carbon dynamics and has become a hotspot for research on the ecosystem’s carbon cycle. Some studies have found that N addition increases the soil organic carbon (SOC) pool [10], but other studies have shown that N addition can aggravate SOC loss [11,12]. The different results depend on the different ecosystems, the level of N application, and the duration of N application. SOC storage increases under N addition [13], especially in decomposed organic horizons in forests [14,15]. There is also a critical value of nitrogen deposition for the promotion of SOC accumulation. The critical N load is estimated to be between 8 and 10 g N m^−2^ year^−1^ [16]. Conversely, the SOC content was reduced following N addition in a *Pinus tabuliformis* forest [17] and Tibetan alpine meadow [18], especially in the 10–20 cm soil layer [19]. Research has found that N addition tends to enhance SOC mineralization [20]. The simulated total SOC response to experimental N addition by a model of soil organic matter (SOM) dynamics showed reduced SOC accumulation under 15 g N·m^−2^ addition [21]. Furthermore, N addition might maintain SOC stability because the soil microbial community adjusts metabolic processes in mesic grasslands [22]. The results reported by Liu et al. suggest that low N addition stabilizes SOC against chemical and biological degradation [23].

The influence of N deposition on the SOC stock depends on the decomposition of the organic carbon input and the dynamic balance of the mineralization process. N deposition has a complex effect on the mineralization of organic carbon by changing the composition and quantity of mineralized substrates and the types and diversity of microorganisms. Recent studies have reported that SOC is linked to soil physical–chemical factors, microbial community composition, and microbiology diversity. First, N addition affects SOM accumulation by changing the soil temperature [24], phosphorus content, and pH gradient [25]. Second, elevated N deposition significantly affects soil microbial abundance, community structure [13,16,26,27], and functional gene activity [28], and the responses of soil microorganisms are strongly correlated with SOC cycling, both directly and indirectly, through multiple soil–microbe interactions. Finally, SOC mineralization is significantly indirectly affected by bacterial diversity [25].

The SOC of grassland ecosystems accounts for 15.5% of the global SOC reserves. The influence of N deposition on SOC in grasslands is an important topic for global climate change research. The *Stipa baicalensis* steppe is an important part of the Eurasian continent grassland and is one of the representative types of temperate grasslands. It is mainly distributed on the SongLiao Plain and the east of the Mongolian Plateau in northern China. Soil microorganisms play a pivotal role in delivering various ecosystem functions and services, but we are only beginning to understand how microbial communities are shaped by various environmental factors and are affected by other soil physical properties. Here, we conducted an N deposition control experiment to explore the responses of SOC concentration to multiple rates of N addition as well as the relationship of soil physical–chemical factors and bacterial community- and nitrogen-regulated SOC in the world’s largest remaining temperate grassland in northern China. We aim to address two questions: (1) What is the effect of N addition on total SOC? and (2) What factors mediate the response of total SOC to N addition in the soil ecosystem? We hypothesize that N addition would influence SOC by mediating soil physical–chemical factors, bacterial diversity, and community structure. 

## 2. Materials and Methods

### 2.1. Site Description

We conducted field investigations in temperate grasslands in northern China. The fields are located in a typical area in the *Stipa baicalensis* steppe (48°30′N, 119°42′E; 765 m) in the Hulun Buir Grassland, Inner Mongolia (Figure 1). The climate is a typical temperate continental monsoon climate, with cool and windy winters, hot summers, and drought and scarcity of rain year-round. The mean annual precipitation is approximately 396 mm. Approximately 66% of the annual precipitation occurs in summer, especially in July. Approximately 16% of the annual precipitation occurs in autumn. The mean annual temperature is −0.7 °C. The *S. baicalensis* meadow steppe, mostly containing Haplic Calcisols (according to the FAO classification), is dominated by *S. baicalensis* and *Leymus chinensis*. The common representative plants are *Achnatherum sibiricum*, *Carex pediformis*, *Filifolium sibiricum*, *Thalictrum petaloideum*, *Melissitus ruthenica*, *Serratula centauroides*, *Cleistogenes squarrosa*, and *Carex duriuscula*. The community is composed of 66 species belonging to 21 families and 49 genera.

### 2.2. Experimental Design and Field Measurements

From 2010, N deposition was simulated using N fertilizer. A randomized block design with 4 replications was adopted. In addition to atmospheric N deposition, N was added at eight levels [29]: 0, 1.5, 3.0, 5.0, 10.0, 15.0, 20.0, and 30.0 g N·m^−2^ yr^−1^, referred to as low N addition (1.5 g N·m^−2^ yr^−1^) [4,5], moderate N addition (3.0, 5.0, and 10.0 g N·m^−2^ yr^−1^), high N addition (15.0, 20.0, and 30.0 g N m^−2^ yr^−1^), and CK (0 g N·m^−2^ yr^−1^) [30] based on atmospheric N deposition data and experimental results. These 8 experimental treatments resulted in 32 plots (8 × 8 m), and the buffer zones were 5 m wide between the plots and 2 m wide between the different treatment plots (illustrated in Appendix A). The test area was enclosed and protected. Nitrogen was added every year in mid-June and mid-July. The nitrogen fertilizer used was NH_4_NO_3_, which was dissolved in water and sprayed evenly in each treatment; the nontreated plots received an equal volume of deionized water. The process of N addition was carried out by repeated uniform spraying to approximately simulate the process of atmospheric N deposition and rainfall.

### 2.3. Sampling and Chemical Analyses

On 10 August 2015, in each of the 32 plots, soil subsamples were collected using a soil corer (0–10 cm deep, 2 cm inner diameter) from 10 random points across each plot and mixed to yield one composite sample per plot. The litter layer was carefully removed before sampling. The soil samples were stored in airtight polypropylene bags and placed in a cool box at 4 °C during transport to the laboratory. Litter, roots, and stones were carefully removed by hand, and the soil was divided into several subsamples. Subsamples for soil physicochemical factor analyses were stored at 4 °C for no longer than 1 week. Subsamples for soil physicochemical factors were air-dried. Subsamples for bacterial community composition and high-throughput sequencing analysis were stored at −80 °C.

### 2.4. Soil Physicochemical Factors

Soil pH was determined at a soil/water ratio of 1:2.5 using a Delta 320 pH meter (Mettler Toledo Instruments, Shanghai, China). The total organic carbon content was measured by dry combustion with a macro elemental analyzer (Vario MAX C/N; Elementar Analysensysteme, Hanau, Germany). For ammonium (NH_4_-N) and nitrate (NO_3_-N) content determination, soils were extracted with 0.5 M K_2_SO_4_ and then examined using an FIA Star 5000 flow-injection autoanalyzer (Foss Tecator, Höganäs, Sweden).

### 2.5. DNA Extraction and Illumina Sequencing

Bulk bacterial community DNA was extracted from each soil sample using the Power Soil DNA isolation kit according to the manufacturer’s instructions. The extracted soil DNA was examined by 1% agarose gel electrophoresis, and an ultra-micro spectrophotometer (NanoDrop 2000, New York, Massachusetts, USA) was used for quality inspection [31]. The Qubit 2.0 A DNA detection kit was used to accurately quantify the extracted DNA to determine the amount of DNA to be added to the PCR. The primers used for amplification of the V3–V4 hypervariable regions of the bacterial 16S rRNA gene were 336F (5′-GTACTCCTACGGGAGGCAGCA-3′) and 806R (5′-GGACTACHVGGGTWTCTAAT-3′). Barcode-specific primers were used to distinguish individual samples. The PCR amplification system was as follows: 10× Pyrobest buffer, 5 μL; dNTPs (2.5 mmol·L^−1^), 4 μL; primer F (10 μmol·L^−1^), 2 μL; primer R (10 μmol·L^−1^), 2 μL; Pyrobest DNA polymerase (2.5 U·L^−1^), 0.3 μL; 30 ng of DNA template; and deionized water to 50 μL. The reaction conditions were as follows: predenaturation at 95 °C for 5 min; 25 cycles of 95 °C for 30 s, 56 °C for 30 s, and 72 °C for 40 s; and extension at 72 °C for 10 min. After the completion of the amplification reaction, agarose gel electrophoresis was performed on the PCR products, and DNA was recovered using the Shanghai Shengong agarose recovery kit (cat: SK8131). The Qubit 2.0 fluorescence quantitative system was used to determine the concentration of the recovered product, and amplicons of equal molar concentrations were pooled together and then mixed for sequencing. For each soil sample, three extractions were performed in parallel. The extracts were PCR amplified, mixed, and submitted for sequencing [32]. A library was constructed, and all sequences were generated with the MiSeq (Illumina) platform (2 × 250 bp) using paired-end reads. All of the abovementioned steps were completed by the Allwegene Tech Company (Beijing, China).

### 2.6. Bioinformatics

Paired-end sequencing was performed on an Illumina platform (MiSeq). Low-quality reads were removed after preprocessing of the data by Trimmomatic software and Readfq software (version 6.0) and then paired reads were spliced into a sequence according to an overlapping relationship between PE data by FLASH software (version 1.2.10). To obtain high-quality reads, the internal program was used to remove barcode sequences and primer sequences at both ends of tags by Mothur software. Clean tags were obtained after removing chimeras and short sequences based on USEARCH (version 8.0.1623).

Merging, quality filtering, and 97% de novo operational taxonomic unit (OTU) clustering were performed using QIIME software (v1.8.0). A total of 147,705 OTUs were generated from 32 samples, among which the number of singleton OTUs (OTUs with an abundance of 1) was 116518. Singleton OTUs may be caused by sequencing errors, so these OTUs were removed. The number of non-singleton OTUs was 31,187.

Data were extracted by random sampling, and a curve was constructed based on the number of sequences extracted and the number of representative OTUs, that is, the dilution curve. The dilution curves of 32 samples are shown (Appendix A). Shannon values of different random samples were calculated by Mothur software (Appendix A). The results showed that the sequencing volume of all samples was sufficient, and most microbial communities could be measured. The species accumulation curve is used to describe the increase in species with increasing sample size. It is an effective tool to investigate the species composition of samples and predict species abundance in samples. The results reflect the rate of emergence of new OTUs (new species) under continuous sampling (Appendix A). Within a certain range, the curve tends to be flat with increasing sample size, indicating that species in this environment will not increase significantly with increasing sample size. The core microbiome is the microbiome that covers all samples. The total OTU number of 32 samples is related to the number of samples (Appendix A). The core microbiome of 32 samples contained a total of 423 OTUs.

### 2.7. Data Processing and Analysis

One-way ANOVAs with Duncan’s multiple range tests were used to determine the significance of differences along the N addition gradient with respect to soil properties (including soil pH, NO_3_-N, NH_4_-N, and TOC), bacterial diversity (including OTU richness, Chao1, goods coverage, observed species, PD whole tree, and Shannon) and the relative abundance of dominant phyla.

Linear regressions were used to assess how soil pH, NO_3_-N, NH_4_-N, and OTU richness were related to SOC. Pearson’s correlation was conducted between soil properties, bacterial diversity, relative abundance of bacterial phyla, and SOC.

We used structural equation modeling (SEM) to estimate the strength of direct and indirect relationships between N addition, soil physicochemical index, bacterial diversity and community structure, and SOC. In the model, we assumed that N addition had the potential to alter total SOC directly, as well as indirectly, by changing soil pH, NO_3_-N, and the bacterial index. We used the chi-square test, Akaike information criterion, and the root mean square error of approximation to evaluate the goodness of fit of the model.

SEM analyses were performed using IBM AMOS 21.0. The remaining statistical analyses were conducted using IBM SPSS Statistics 20.0.

## 3. Results

### 3.1. Effect of Nitrogen Deposition on Soil Properties

Over the 6-year experimental period, N addition enhanced the SOC content, with an increase of 9.04 g·kg^−1^ (22.8%) compared to CK under high N addition (*p* < 0.01; Figure 2 and Appendix A). However, the enhancement in the SOC content induced by N addition was not significant in the low and moderate N addition treatments.

N addition decreased soil pH. Compared to CK, the soil pH decreased by 7.5% and 15.8% under moderate N addition and high N addition (*p* < 0.001; Figure 3a and Appendix A) but showed no significant variation under low N addition (Figure 3a). Compared to CK, high N addition significantly enhanced NO_3_-N by 37.4 mg·kg^−1^ (575.7%) (*p* < 0.01; Figure 3b and Appendix A). However, the enhancement in NO_3_-N was not significant in the low and moderate N addition treatments. The NH_4_-N increased by 21.3 mg·kg^−1^ (76.0%) under moderate N addition compared to that in CK (*p* < 0.001; Figure 3c and Appendix A) but did not change under low and high N addition.

### 3.2. Effect of Nitrogen Deposition on Bacteria

The OTU richness was highest under low nitrogen addition. Compared to CK, low N addition enhanced the OTU richness by 23.0% (*p*< 0.01; Figure 4 and Appendix A). However, the variation in the OTU richness enhanced by N addition was not significant in the moderate and high N addition treatments. Compared to low N addition, moderate N addition and high N addition significantly decreased the OTU richness by 20.6% and 23.5%, respectively (*p* < 0.01; Figure 4 and Appendix A). The effect of N addition on other diversity indices (including Chao1, goods coverage, observed species, PD whole tree, and Shannon index) was not significant (*p* > 0.05; Appendix A).

After quality trimming and chimera removal, the bacterial sequences were clustered into 31,187 OTUs using a Bayesian classifier at the 97% similarity level. The relative abundance of the dominant bacterial phyla changed. The soil bacterial community was dominated by Acidobacteria (27.3%), followed by Actinobacteria (12.0%), Proteobacteria (13.2%), Chloroflexi (5.0%), Verrucomicrobia (6.3%), and Gemmatimonadetes (1.3%). When all the dominant phyla with a relative abundance of >1% were considered, a significant difference was observed in the phylum Proteobacteria among the different N addition treatments (*p* < 0.01; Appendix A). Compared to CK, moderate and high N addition significantly enhanced the relative abundance of Proteobacteria (from 7.6% under CK to 17.2% under high N addition). Proteobacteria became the second dominant phylum after Acidobacteria under high N addition.

### 3.3. Factors Affecting Soil Organic Carbon

Regression analysis showed that the SOC content was negatively correlated with bacterial OTU richness (*r*^2^ = 0.368, *p* = 0.038, Figure 5a and Appendix A) and pH (*r*^2^ = 0.809, *p* < 0.001, Figure 5b and Appendix A) but positively correlated with NO_3_-N (*r*^2^ = 0.897, *p* < 0.001, Figure 5c and Appendix A). In contrast, no significant relationship was found between SOC and NH_4_-N (*p* > 0.05, Figure 5d and Appendix A).

SEM explained 56% of the variation in the SOC content (Figure 6). The SEM results (Appendix A, Figure 6) showed that N addition affected the SOC content by changing the pH and the bacterial community composition. The relationship between the relative abundance of Proteobacteria and SOC was positive. In contrast, the relationship between the relative abundance of Planctomycetes and SOC was negative under N addition. However, the contribution of bacterial diversity was limited.

## 4. Discussion

We aimed to explore the responses of the SOC content to N addition through a 6-year N treatment applied to a *S. baicalensis* steppe. Three main results emerged: (1) high N addition enhanced the SOC content; (2) the SOC content was associated with soil NO_3_-N concentration, pH, and bacterial diversity; and (3) N addition enhanced SOC, which may be mediated by the effects of N on bacterial community structure.

Research findings on N enrichment resulting in a shift in organic carbon have been inconsistent. On the one hand, N addition is thought to enhance the SOC content. Cheng et al. [16] found that N input enhanced the SOC content by altering the microbial community. This finding is in agreement with our results. On the other hand, N addition has been found to reduce the SOC content and increase the decomposition of SOC vis changes in belowground chemical properties and aboveground plants in a Tibetan alpine meadow [18]. The SOC concentration in the 0–10 cm soil layer showed no response to N addition in a semiarid grassland [19]. The different response patterns between ecosystems with different soil properties and microbial community structures may be ecosystem-dependent.

Several mechanisms may potentially explain the observed changes in the SOC content. First, N can influence soil properties. For instance, Li et al. [24] found that changes in microbiology composition were not correlated with the SOC content, but temperature affected SOM decomposition. N deposition causes soil acidification and affects the SOC content [22]. Second, the microbial community plays an important role in this process of N addition, affecting SOC accumulation. Several recent studies have highlighted the important role of the bacterial community in determining the SOC content [13,26,33]. The SYMPHONY model also indicated that atmospheric N deposition may promote SOC accumulation via changes in the structure and metabolic activities of microbial communities [34]. Finally, the impact of microbial diversity on SOC is also very important. SOC mineralization in red soil is controlled by microbial diversity [25]. The decomposition process of SOM influences microbial diversity.

In the present study, we suggest that the bacterial community structure could affect the SOC under N deposition. Elevated chronic N deposition significantly affects the soil microbial community structure [28]. N additions increase soil acidification. Soil acidification can have direct effects on the microbial community composition. N addition changes microbial degradation, alters the molecular composition of SOM, and enhances soil carbon storage. Bacterial community structure was reported to be related to SOC decomposition under high N addition [13,16], indicating the key mechanisms driving SOC accumulation. Under high N addition, soil acidification and the available N content increased, and the dominance of a few bacteria, such as Proteobacteria, increased in the bacterial community. The key functional gene groups responsible for C degradation are related to slower SOC decomposition, revealing the pivotal mechanisms driving SOC accumulation under high N addition [13]. In our results, the bacterial diversity indices, such as OTUs, fluctuated with increasing N, and the contribution of bacterial diversity to the SOC content was very limited. In contrast to this result, Naveed et al. [33] found that the ratio of clay to organic carbon was significantly correlated with bacterial richness and diversity indices. This difference may be explained by the different ecosystems and soil environments. Overall, soil properties and bacterial community structure have an impact on SOC under N deposition. 

Our results indicate that N addition increased SOC, which may be associated with soil N availability, pH value, and bacterial community structure. These results indicate that environmental conditions under high N concentrations may benefit the survival of Proteobacteria (Appendix A). Proteobacteria had significant positive effects on the SOC fraction [35]. Planctomycetes were negatively correlated with the SOC content under N addition (Appendix A). Specific groups of aerobic Planctomycetes participate in degrading organic matter [36]. Other research found that SOC was the main driving factor changing the bacterial communities [37]. Therefore, there is a complex interactive relationship between SOC and microbes.

In conclusion, N addition altered the SOC content, which may be mediated by the effects of N on the soil physicochemical properties and bacterial community structure. The effects of soil bacterial diversity on SOC might be overestimated in N addition experiments.

## Figures and Tables

**Figure 1 microorganisms-08-00326-f001:**
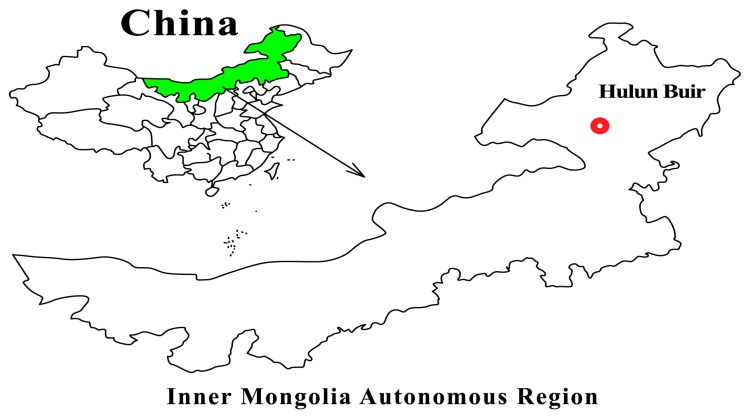
Geographic locations (red dots) of study sites in northern China.

**Figure 2 microorganisms-08-00326-f002:**
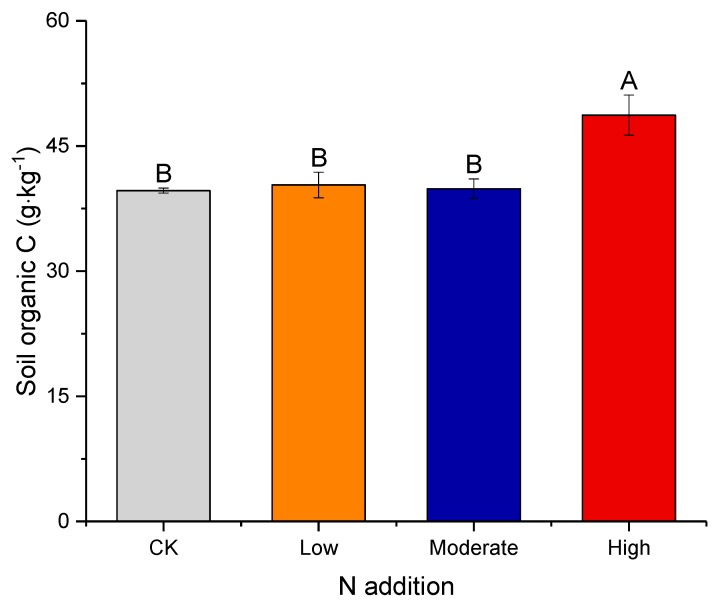
Effects of N addition on soil organic carbon. Error bars show one standard error of the mean. Different letters above bars indicate significant differences according to Duncan’s multiple range test (*p* < 0.05).

**Figure 3 microorganisms-08-00326-f003:**
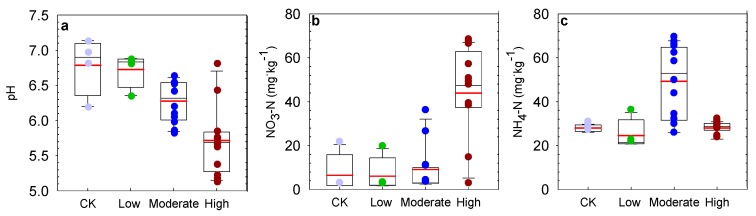
Effects of N addition on pH (**a**), NO^3^-N (**b**), and NH^4^-N (**c**). Error bars show one standard error of the mean.

**Figure 4 microorganisms-08-00326-f004:**
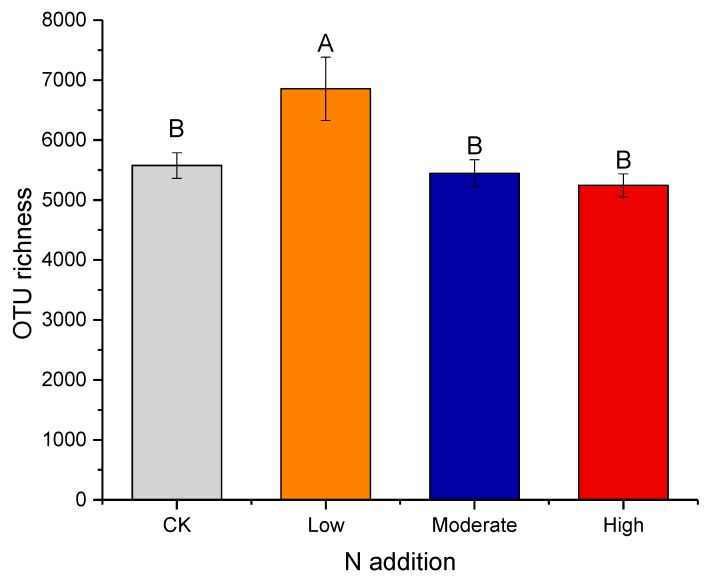
Effects of N addition on bacterial OTU richness. Error bars show one standard error of the mean. Details are as in Figure 2.

**Figure 5 microorganisms-08-00326-f005:**
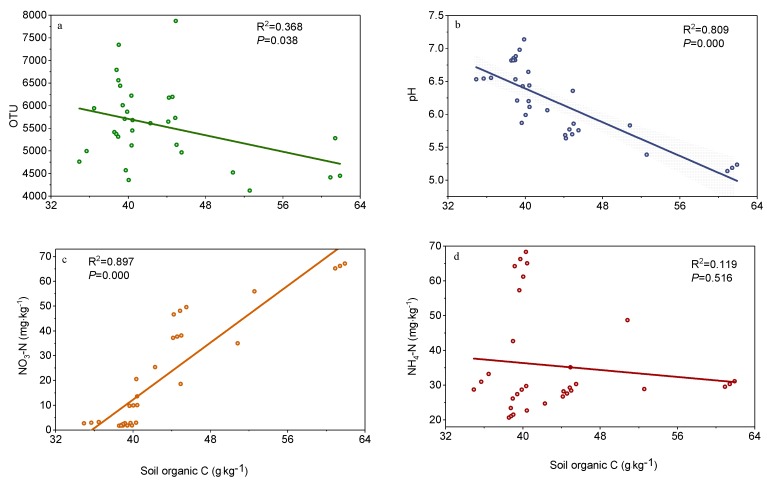
Linear regressions of soil organic carbon against (**a**) OTU, (**b**) soil pH, (**c**) NO_3_-N, and (**d**) NH_4_-N.

**Figure 6 microorganisms-08-00326-f006:**
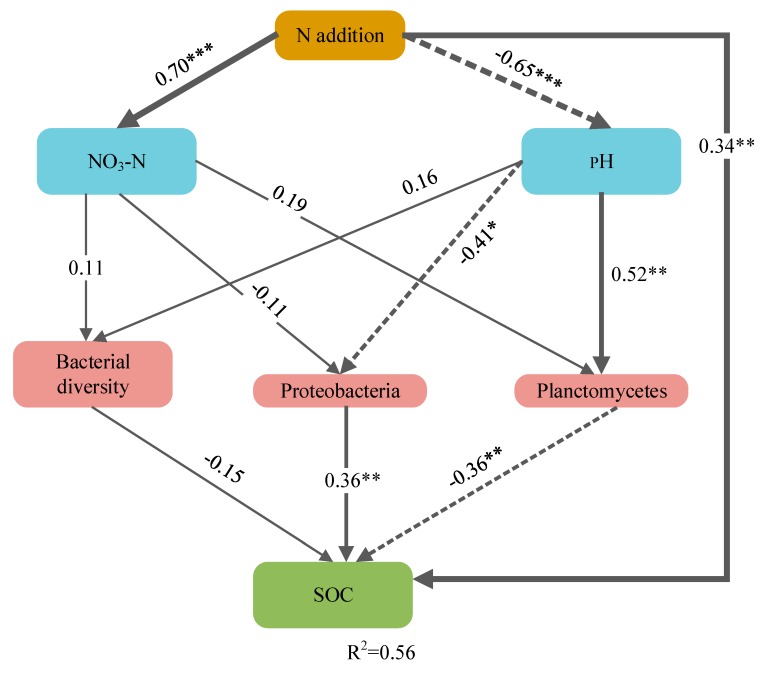
Structural equation modeling of N addition, pH, NO^3^-N, bacterial diversity (OTU richness), and the relative abundance of Planctomycetes and Proteobacteria against soil organic carbon (SOC) content. Numbers adjacent to arrows are standardized path coefficients and indicate the effect size of the relationship. R^2^ represents the proportion of variance in each dependent variable explained by the model. Arrows indicate positive (solid) and negative (dashed) relationships. The arrow width is proportional to the strength of the relationship. Goodness-of-fit statistics for each model are shown below the model. *** *p* ≤ 0.001, ** *p* ≤ 0.01, * *p* ≤ 0.05. X^2^ = 10.151, df = 9, *p* = 0.338; RMSEA = 0.064; GFI = 0.912; AIC = 48.150.

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
