# Peer review of "The Roles of Bacteria in Soil Organic Carbon Accumulation under Nitrogen Deposition in Stipa baicalensis Steppe"

_microorganisms, 2020, doi:10.3390/microorganisms8030326_

Round 1
Reviewer 1 Report
Check the style of the references in the text. There are some misprint I guess Check the acronyms, they are not always explained In some part of the text you should improve English, the concept is not always well understandable I suggest to insert a scheme of the experimental design. The paragraph 2.2 is not well understandable. How the different levels of N are "located" in the field? I think that this part should be described better. Does the plot with 0 addition of N should be used as blank instead of a point of the "low N addition level"? I think that is better to include the amount of soil used for DNA extraction in paragraph 2.5. In the same way a more detailed description of samples and libraries preparation, would be preferable The assertion at line 267 of the text is true only for the Proteobacteria I do not agree with the statement at line 301. I don't see a real shift in the microbial community. Even if the relative abundance of Proteobacteria increase from 6.9% to 17.2% and Actinobacteria decrease fron 33% to 24%, Acidobacteria remain the main phyla in all the conditions. Chloroflexi, Verrucomicrobia, Gemnatimonadetes seem to remain almost the same. There is a change but not a shift. A discussion with all the relative abundances should be useful. In addition, I see that there is an extensive statistical data processing but a comparison between the community structure at the beginning of the study and at the end should be more useful in my opinion. Also the interaction between the flora described and the strains should be investigated
Figure:
Fig. 1 has not a good resolution
Fig 2 and 4. I think that a more detailed description of the data is necessary
Author Response
Firstly, we would like to thank you for the constructive comments concerning our article. These comments are all valuable and helpful for improving our article. It is with excitement that I resubmit to you a revised version of the manuscript. I have responded specifically to each suggestion below, beginning with your own. To make the changes easier to identify where necessary, I have numbered them. All the authors have seriously discussed all these comments. In this revised version, changes to our manuscript within the document were all highlighted by using red-colored text. Point-by-point responses to the reviewers are listed below in this letter.
And, please set the manuscript to Simple display of the revision status of the mark of the manuscript (the number of lines is different in the different station)
1)Check the style of the references in the text.
Response 1: Thanks for your nice suggestions. We have formatted all the references according to the style of Microorganisms, and edited the references using endnote software and mdpi.ens, and double-checking
2)There are some misprint I guess Check the acronyms, they are not always explained
Response 2: Thanks for reminding, I have revised the text of the manuscript, I check the acronyms problem again
3)In some part of the text you should improve English, the concept is not always well understandable
Response 3: Thanks for the advice, the manuscript has been revised many times and communicated with language editing experts to write the smoothest language by AJE (American Journal Experts) language company.
4)I suggest to insert a scheme of the experimental design.
Response 4: thanks for the advice, I already add a figure on scheme of the experimental design (figure S1)
5)The paragraph 2.2 is not well understandable. How the different levels of N are "located" in the field? I think that this part should be described better. Does the plot with 0 addition of N should be used as blank instead of a point of the "low N addition level"?
Response 5: N was added at eight levels(Stevens, Dise et al. 2004), and the N addition gradient was based on the previous research, that Wang et al defined the low N addition(3.5 g N·m-2), moderate N addition (7.0 g N·m-2) and high N addition (14.0 g N·m-2)(Wang, Bao et al. 2015). And when N addition beyond the 15 g N·m-2·, the effect of nitrogen addition is negative (Tonitto, Goodale et al. 2014), so more than 15 nitrogen additions are considered excessive(high N addition) . we add more details information in paragraph 2.2 .
Thanks for the advice, we change the N addition, and 0 was CK. In this regard, the data analysis and figure(figure 2, 3 and 4 figure S6 and table S1) are also re-performed. We also rewrite the result part about the change.
6)I think that is better to include the amount of soil used for DNA extraction in paragraph 2.5. In the same way a more detailed description of samples and libraries preparation, would be preferable
Response 6: good advice. We add more details information in paragraph 2.5. It contains Primers, PCR amplification system, and reaction conditions, etc. (Line 166-187).
7)The assertion at Line 267 of the text is true only for the Proteobacteria I do not agree with the statement at Line 301. I don't see a real shift in the microbial community. Even if the relative abundance of Proteobacteria increase from 6.9% to 17.2% and Actinobacteria decrease fron 33% to 24%, Acidobacteria remain the main phyla in all the conditions. Chloroflexi, Verrucomicrobia, Gemnatimonadetes seem to remain almost the same. There is a change but not a shift. A discussion with all the relative abundances should be useful. In addition, I see that there is an extensive statistical data processing but a comparison between the community structure at the beginning of the study and at the end should be more useful in my opinion. Also the interaction between the flora described and the strains should be investigated
Response 7: thank for the sentence that “There is a change but not a shift”. We describe the result of figure S6. Some dominant bacteria have changed their relative abundance, but the more important thing is the “shift” was no significant. We found the important information that Proteobacteria became the second most dominant phylum under high N addition. (Line 272-276)
At the same time, we found the discussion section also have unclear sentence, like the bacteria community change. We revised the 3,4 and 5 paragraph of discussion. (Line 309-352)
We think the interaction between the flora described and the strains is very important for this research. We fully agree with your point. However, soil microorganisms are difficult to cultivate, and isolating strains from complex microorganisms in the soil is a challenge. In our other research, we study the function microorganisms, like carbon-fixing bacteria (cbbL), nitrogen-fixing microorganisms (nifH) and ammonia-oxidizing microorganisms (AOA, AOB). We think the next step study will interesting and meaningful.
Figure:
8)Fig. 1 has not a good resolution
Response 8: we were really sorry for your careless mistakes. Thank you for your reminding. we replaced the original figure with a high-resolution figure (figure 1)
9)Fig 2 and 4. I think that a more detailed description of the data is necessary
Response 9: thank for the advice, we add a more detailed description of the result in figure 2 and figure 4. (Line 244-247, Line 259-263)
Thank you again for your positive comments and valuable suggestions to improve the quality of our manuscript. If there are any other modifications we could make, we would like very much to modify them and we really appreciate your help. Thank you very much for your help.
Reference:
Stevens, C. J., N. B. Dise, J. O. Mountford and D. J. Gowing (2004). "Impact of nitrogen deposition on the species richness of grasslands." Science 303(5665): 1876-1879.
Tonitto, C., C. L. Goodale, M. S. Weiss, S. D. Frey and S. V. Ollinger (2014). "The effect of nitrogen addition on soil organic matter dynamics: a model analysis of the Harvard Forest Chronic Nitrogen Amendment Study and soil carbon response to anthropogenic N deposition." Biogeochemistry 117(2-3): 431-454.
Wang, J., J. T. Bao, J. Q. Su, X. R. Li, G. X. Chen and X. F. Ma (2015). "Impact of inorganic nitrogen additions on microbes in biological soil crusts." Soil Biology & Biochemistry 88: 303-313.
Reviewer 2 Report
The subject of the paper is of interest to a wide audience. Experiments are clearly described. However, there a few drawbacks that render the manuscript not ready for publication in the present form.
1) The Introduction section should be shortened and much more concise, not just a mere inordinate list of relevant publications. The objective of the present work should be stated more clearly in order to better evaluate and appreciate the originality of this work.
2) Primers used to track microbial populations are compatible also with archaea. Did the authors find them in their samples?
3) In light of the results presented in the paper, speculations in the Discussion section are overwhelming and not always warranted by the results. The Discussion of the results should, therefore, be widely shortened. According to my opinion, there is confusion in the interpretation of statistical analyses. These show an association between environmental factors and microbial population dynamics and not a causal relationship as suggested by authors. Authors should be careful in their statements. Moreover, microbial data are presented in a contradictory way when analyzed in relation to N deposition and SOC changes (lines 254-271).
4) In respect of the contradictory results of previous papers on the same subject, it is not clear which are the main conclusions of the present work. I suggest to omit the Conclusion section and to add no more than two sentences as a conclusion at the end of the Discussion section.
Author Response
Firstly, we would like to thank you for the constructive comments concerning our article. Thank you again for your positive comments and valuable suggestions to improve the quality of our manuscript. These comments are all valuable and helpful for improving our article. All the authors have seriously discussed all these comments. According to the reviewers’ comments, we have tried best to modify our manuscript to meet the requirements. In this revised version, changes to our manuscript within the document were all highlighted by using red-colored text. Point-by-point responses to the reviewers are listed below in this letter.
And, please set the manuscript to Simple display of the revision status of mark of the manuscript (the number of lines is different in different station)
1)The Introduction section should be shortened and much more concise, not just a mere inordinate list of relevant publications. The objective of the present work should be stated more clearly in order to better evaluate and appreciate the originality of this work.
Response 1: thanks for the advice, we reorganized the introduction, especially the 2 paragraph of the Introduction section. We briefly present the results of other studies. (Line 68-80)
2) Primers used to track microbial populations are compatible also with archaea. Did the authors find them in their samples?
Response 2: good question. The archaea are also used in the 16S V3-V4 area of the primer, but the primer sequence is different. 1) The soil we studied contained a large number of archaea (we analyzed the ammonia oxidation archaea(aomA), found that its abundance is about 4.7×108copies·g-1 soil, there is about 72 OTUs); 2) from the sequencing results did not find the archaea (of which 20% of the relative abundance of bacteria is unconfirmed, there may be archaea. This has not yet been determined, further experimental analysis is required); 3) the N deposition was simulated from 2010, consulting other experts, the results of the study that the 16S RNA detected the vast majority of bacteria, may also contain very small amounts of archaea.
3)In light of the results presented in the paper, speculations in the Discussion section are overwhelming and not always warranted by the results. The Discussion of the results should, therefore, be widely shortened. According to my opinion, there is confusion in the interpretation of statistical analyses. These show an association between environmental factors and microbial population dynamics and not a causal relationship as suggested by authors. Authors should be careful in their statements. Moreover, microbial data are presented in a contradictory way when analyzed in relation to N deposition and SOC changes (Lines 254-271).
Response 3: thank you for the advice. We're aware of one problem that a real “shift” in the microbial community. Even if the relative abundance of dominant phylum changed, but the dominating phylum have not shifted, so there is a change but not a shift. The next problem is an inaccurate causal relationship. The dominant bacteria played an important role in increasing the SOC content, but not the dominant bacteria lead to the increase of the SOC content. For that, we modified the result and discussion, especially language statements. (Line 321-331; Line 342-352)
We agree with the contradiction of microbial data. we were really sorry for our careless mistakes. Thank you for your reminding. We check the data analysis, and found a mistake in the structural equation model. We reanalyze the date and redraw the figure (figure 6).
4) In respect of the contradictory results of previous papers on the same subject, it is not clear which are the main conclusions of the present work. I suggest to omit the Conclusion section and to add no more than two sentences as a conclusion at the end of the Discussion section.
Response 4: thank you for the advice. We omitted the Conclusion section and to add two sentences as a conclusion at the end of the Discussion section. (Line 354-357)
Thank you again for your positive comments and valuable suggestions. If there are any other modifications we could make, we would like very much to modify them and we really appreciate your help. Thank you very much for your help.
Round 2
Reviewer 2 Report
1) The English language needs absolutely profound editing.
2) The aim of the work is not clearly stated (lines 129-130). The two questions are very, very similar.
3) As I told in the first revision, caution regarding the role of microorganisms is needed. The authors should correct accordingly their statements.
Author Response
We would like to thank you for constructive comments concerning our article. These comments are all valuable and helpful for improving our article. It is with excitement that I resubmit the revised version of manuscript. I have responded specifically to each suggestion below, beginning with your own. All the authors have seriously discussed about all these comments. In this revised version, changes to our manuscript within the document were highlighted by using red coloured text. Point-by-point responses to the reviewers are listed below this letter.
1) The English language needs absolutely profound editing.
Response 1: We thank the reviewer for the comments. We have edited the revised manuscript many times through the language company and required more professional editors to get a smoother language. Attachment is the language editing certificate for the manuscript.
2) The aim of the work is not clearly stated (lines 129-130). The two questions are very, very similar.
Response 2: We thank the reviewer for the comments. As suggested, we now rewrite the aim of the research and the scientific question. (line 103-107)
3) As I told in the first revision, caution regarding the role of microorganisms is needed. The authors should correct accordingly their statements.
Response 3: We thank the reviewer for the comments. As suggested, we now rewrite the sentences of the revised manuscript by the red highlight. (line 298-299,323-324, 338-339, 341-341,345,350-351).
Thank you again for your valuable suggestions to improve the quality of our manuscript. If there are any other modifications we could make, we would like very much to modify them and we really appreciate your help. Thank you very much for your help.
